# Coconut Oil Alleviates the Oxidative Stress-Mediated Inflammatory Response via Regulating the MAPK Pathway in Particulate Matter-Stimulated Alveolar Macrophages

**DOI:** 10.3390/molecules27092898

**Published:** 2022-05-02

**Authors:** Xinyu Chen, Dong Im Kim, Hi-Gyu Moon, Minchul Chu, Kyuhong Lee

**Affiliations:** 1Inhalation Toxicology Center for Airborne Risk Factor, Korea Institute of Toxicology, 30 Baehak1-gil, Jeongeup-si 56212, Korea; chen.xinyu@kitox.re.kr (X.C.); dongim.kim@kitox.re.kr (D.I.K.); higyu.moon@kitox.re.kr (H.-G.M.); 2Department of Human and Environmental Toxicology, University of Science & Technology, Daejeon 34113, Korea; 3Greensol Co., Ltd., 89-26, Jimok-ro, Paju-si 10880, Korea; chumin6329@naver.com

**Keywords:** artificial particulate matter, diesel exhaust particulates, coconut oil, oxidative stress, inflammation, mitogen-activated protein kinase, alveolar macrophage

## Abstract

Exposure to particulate matter (PM) is related to various respiratory diseases, and this affects the respiratory immune system. Alveolar macrophages (AMs), which are defenders against pathogens, play a key role in respiratory inflammation through cytokine production and cellular interactions. Coconut oil demonstrates antioxidant and anti-inflammatory properties, and it is consumed worldwide for improved health. However, reports on the protective effects of coconut oil on the PM-induced respiratory immune system, especially in AMs, are limited. In this study, we generated artificial PM (APM) with a diameter approximately of 30 nm by controlling the temperature, and compared its cytotoxicity with diesel exhaust particles (DEP). We also investigated the antioxidant and anti-inflammatory effects of coconut oil in APM– and DEP–stimulated AMs, and the underlying molecular mechanisms. Our results showed that APM and DEP had high cytotoxicity in a dose-dependent manner in AMs. In particular, APM or DEP at 100 μg/mL significantly decreased cell viability (*p* < 0.05) and significantly increased oxidative stress markers such as reactive oxygen species (*p* < 0.01); the GSSH/GSH ratio (*p* < 0.01); and cytokine production, such as tumor necrosis factor-α (*p* < 0.001), interleukin (IL)-1β (*p* < 0.001), and IL-6 (*p* < 0.001). The expression of the genes for chemokine (C-X-C motif) ligand-1 (*p* < 0.05) and monocyte chemoattractant protein-1 (*p* < 0.001); and the proteins toll-like receptor (TLR) 4 (*p* < 0.01), mitogen-activated protein kinase (MAPK), and c-Jun N-terminal kinase (*p* < 0.001), p38 (*p* < 0.001); and extracellular receptor-activated kinase (*p* < 0.001), were also upregulated by PM. These parameters were reversed upon treatment with coconut oil in APM– or DEP–stimulated AMs. In conclusion, coconut oil can reduce APM– or DEP–induced inflammation by regulating the TLR4/MAPK pathway in AMs, and it may protect against adverse respiratory effects caused by PM exposure.

Exposure to particulate matter (PM) is related to various respiratory diseases, and this affects the respiratory immune system. Alveolar macrophages (AMs), which are defenders against pathogens, play a key role in respiratory inflammation through cytokine production and cellular interactions. Coconut oil demonstrates antioxidant and anti-inflammatory properties, and it is consumed worldwide for improved health. However, reports on the protective effects of coconut oil on the PM-induced respiratory immune system, especially in AMs, are limited. In this study, we generated artificial PM (APM) with a diameter approximately of 30 nm by controlling the temperature, and compared its cytotoxicity with diesel exhaust particles (DEP). We also investigated the antioxidant and anti-inflammatory effects of coconut oil in APM– and DEP–stimulated AMs, and the underlying molecular mechanisms. Our results showed that APM and DEP had high cytotoxicity in a dose-dependent manner in AMs. In particular, APM or DEP at 100 μg/mL significantly decreased cell viability (*p* < 0.05) and significantly increased oxidative stress markers such as reactive oxygen species (*p* < 0.01); the GSSH/GSH ratio (*p* < 0.01); and cytokine production, such as tumor necrosis factor-α (*p* < 0.001), interleukin (IL)-1β (*p* < 0.001), and IL-6 (*p* < 0.001). The expression of the genes for chemokine (C-X-C motif) ligand-1 (*p* < 0.05) and monocyte chemoattractant protein-1 (*p* < 0.001); and the proteins toll-like receptor (TLR) 4 (*p* < 0.01), mitogen-activated protein kinase (MAPK), and c-Jun N-terminal kinase (*p* < 0.001), p38 (*p* < 0.001); and extracellular receptor-activated kinase (*p* < 0.001), were also upregulated by PM. These parameters were reversed upon treatment with coconut oil in APM– or DEP–stimulated AMs. In conclusion, coconut oil can reduce APM– or DEP–induced inflammation by regulating the TLR4/MAPK pathway in AMs, and it may protect against adverse respiratory effects caused by PM exposure.

## 1. Introduction

Particulate matter (PM), which is a mixture of solid and liquid particles suspended in the air, is an important component of air pollution [1]. Currently, the major sources of environmental PM include vehicle and industrial emissions, waste incineration, power plants, and geological materials [2,3]. PM can be divided into PM 1 (aerodynamic diameter < 1 µm), PM 2.5 (aerodynamic diameter < 2.5 µm), and PM 10 (aerodynamic diameter < 10 µm) [4,5], depending on the particle size. According to the Global Burden of Disease assessment, approximately 4.24 million premature deaths worldwide in 2015 could be attributed to exposure to PM [6]. Toxicological and epidemiological studies have shown that PM exposure can cause human diseases, such as respiratory and cardiovascular diseases [7], since these small particles can enter the respiratory tract and circulate into blood vessels or capillaries [8]. In recent years, increasing attention has been paid to the relationship between PM and public health hazards. Researchers have gradually focused on natural compounds that protect against PM, to boost or strengthen the healthy immune system after PM exposure [9,10].

Macrophages play an important role in the innate immune system [11], and they participate in host defense by engulfing and digesting pathogens [12]. In the respiratory system, alveolar macrophages (AMs) maintain clear air spaces by engulfing foreign materials and minimize the exposure of other airway cells to these substances [13]. Once inhaled and deposited into the lung, PM-pigmented AMs can induce oxidative stress and local inflammation by releasing mediators such as interleukin (IL)-6, IL-8, and tumor necrosis factor (TNF)-α [14] via the activation of stress kinases, such as mitogen-activated protein kinase (MAPK), which is a key signaling pathway that regulates a wide variety of cellular processes, such as proliferation, differentiation, apoptosis, and stress responses under both normal and pathological conditions [15,16,17]. Similar to these observations, our previous study showed that diesel exhaust particulates (DEP) induce endoplasmic reticulum (ER) stress-mediated neutrophil dominant lung inflammation through increasing ER stress markers such as C/EBP homologous protein, and binding immunoglobulin protein, TNF-α, and IL-1β in bronchoalveolar lavage fluids of the DEP–induced murine model and in DEP–stimulated AMs. We also found that AMs play a crucial role in the lung disease process through observing lung inflammation accompanied with the infiltration of DEP–pigment AMs in hematoxylin and eosin-stained lung tissue [18]. Additionally, PM can induce M2 macrophage polarization-enhancing lung diseases. Yan Jiang et al. have reported that PM2.5 induced airway inflammation, compromised lung function, emphysematous lesions, and deleterious airway remodeling in chronic obstructive pulmonary disease (COPD) through inducing the M2 polarization of AMs and upregulating transforming growth factor-β, matrix metallopeptidase (MMP)-9, and MMP-12 [19]. Currently, there is a growing interest in the potential roles of macrophages in the progression of lung diseases such as asthma, COPD, lung fibrosis, and cancer [11,12,19,20].

Coconut is derived from *Cocos nucifera* L., and the coconut oil (65–75%) obtained [21] is widely used in the food and other industries [22,23]. Traditionally, coconut oil has been used as a healthy food, and it has also been utilized for cosmeceutical purposes [24]. Previous studies have reported on the benefits of coconut oil, including several biological activities *in vivo*, such as antioxidant, anti-inflammatory, anticancer, analgesic, antimicrobial, and antipyretic properties [25,26,27]. Coconut oil is considered a saturated fat because it contains more than 90% saturated fatty acids; it also contains high amounts of medium-chain triacylglycerols, which are effective against the development of cardiovascular and inflammatory diseases [28,29], as well as antioxidant compounds. Luiz Henrique C. Vasconcelos [30] demonstrated that coconut oil supplementation reverses oxidative stress-induced peribronchial inflammatory infiltrate, epithelial hyperplasia, smooth muscle thickening, and hypercontractility, through its interactions with the nitric oxide (NO) pathway in guinea pigs. Although coconut oil is a potential candidate for adjuvant therapy with several biological activities, such as antioxidant and anti-inflammatory activities, to date, there are no reports on the anti-inflammatory and antioxidant activities of coconut oil in PM-stimulated AMs. Thus, we established an *in vitro* model of PM-stimulated AMs with inflammatory and oxidative stress responses by PM exposure and investigated the effect of coconut oil in an *in vitro* model.

In the present study, we generated artificial PM (APM) in the laboratory [31] and compared the toxicity of AMs with APM and DEP. In addition, we assessed the protective effects of coconut oil on PM-induced oxidative stress and inflammatory responses in AMs, and investigated its mechanism of action.

## 2. Materials and Methods 

### 2.1. APM Generation

Graphite was selected to produce APM generated from a high-voltage-based soot generator (DNP Digital 3000, Palas GmbH, Germany). It operates on the principle that a spark discharge produces nanoscale particles with consistent concentrations and a high yield. The generator maintains a high voltage of 2500 V and 700 Hz between the two graphite electrodes, using co-mixed gases (1:2) with pure argon (99.999%; 5 L/min) and air (99.999%; 10 L/min). The use of these base gases yielded pure black without volatile impurities. The jump spark rips large amounts of graphite material from the electrodes at high temperatures (Lindberg/Blue M™ 1200 °C split-hinge tube furnaces, Thermo Scientific, Waltham, MA, USA). The APM was sampled using Teflon filters with a pore size of 2 µm and a diameter of 47 mm (R2PJ047, PALL Corporation, Port Washington, NY, USA). A high concentration can result in the coagulation of these nanoparticles into agglomerates. The generated aerosol distribution was quite similar to the distribution of diesel soot particles from the furnace process [31].

### 2.2. Preparation of APM, DEP, and Coconut Oil

Stock suspensions with a concentration of 2 mg/mL of APM and DEP (SRM 2975; National Institute of Standards and Technology, Gaithersburg, MD, USA) were prepared in the growth medium. The suspensions were ultrasonicated using an ultrasonic cleaner (40 kHz, Branson, 1800 Cleaner, Bransonic; St. Louis, MO, USA) for 30 min and an ultrasonicate (20 kHz, VC 505, Vibra-Cell™ Ultrasonic Liquid Processors, Sonics & Materials, Inc., Newtown, CT, USA) for 30 min. Surfactant-free coconut oil in distilled water (DW) emulsion, donated by Dr. Mincheol Chu from GREENSOL Co., Ltd., was prepared in 0.5% coconut oil using high-speed agitation, high pressure, and ultrasonic dispersion methods [32]. Coconut oil has the smallest particle size (0.07~0.3 μm) and a uniform distribution, and it was the most stable during a 7 days stability test during the emulsification process [32].

### 2.3. Measurement of Organic Carbon/Elemental Carbon (OC/EC)

A series of sucrose solutions were used to calibrate the equipment prior to use (N = 5, R^2^ = 1.00). Quartz fiber filters (QMA filter 25 mm, Whatman; Little Chalfont, Buckinghamshire, UK) were pre-baked at 650 °C for 8 h to eliminate organic pollutants and stored in baked aluminum foil before sampling. A 1 cm^2^ punch was removed from the quartz filter after sampling, to analyze the carbonaceous components in APM/DEP using an OC-EC Aerosol Analyzer Model 5 (Sunset Laboratory, Beaverton, OR, USA) according to the revision 8.054 reference method. To avoid the premature evolution of light-absorbing carbon in the OC and EC, the IMPROVE-A temperature protocol was adopted for the OC and EC analysis [33]. 

### 2.4. Cell Culture and APM, DEP, Coconut Oil, and N-Acetylcysteine (NAC) Treatments

A murine alveolar macrophage (MH-S) cell line was obtained from the American Type Culture Collection (Manassas, VA, USA). Cells were maintained in a Roswell Park Memorial Institute (RPMI) 1640 medium supplemented with 10% fetal bovine serum and 1% penicillin–streptomycin solution (Gibco, Grand Island, NY, USA) at 37 °C in a humidified atmosphere of 5% CO_2_. To optimize the treatment concentration, cells were treated with varying concentrations of APM and DEP (50, 100, 250, 500, 1000, and 2000 μg/mL), coconut oil (0.005, 0.0125, 0.025, and 0.05%), and NAC (1, 5, 10, 15, and 20 mM). In addition, cells were treated with DW as a vehicle control for coconut oil, and cytotoxicity was evaluated. Next, to investigate the protective effect of coconut oil in PM-stimulated MH-S, cells were pretreated with coconut oil at 0.0125 or 0.025% (*v*/*v*) 6 h before adding APM or DEP (100 μg/mL); 6 h later, the cells were washed twice with Dulbecco’s phosphate-buffered saline (PBS) and harvested. The cells were treated with antioxidant NAC (10 mM) for 2 h (Sigma-Aldrich, St. Louis, MO, USA).

### 2.5. Cell Viability Assay

The cytotoxicity of APM, DEP, and coconut oil was assessed using the 3–(4,5–dimethylthiazol–2–yl)–2,5–diphenyltetrazolium bromide (MTT; Sigma-Aldrich) assay [34]. After the coconut oil, APM, and DEP treatments, the cells were incubated with 100 μL of 1 mg/mL MTT for 3 h at 37 °C. The medium was discarded, and 100 μL dimethyl sulfoxide (Junsei Chemical Co.; Tokyo, Japan) was added to each well to dissolve the formazan crystals. Absorbance was measured on a microplate reader (BioTek; Synergy Mx, Winooski, VT, USA) at 570 nm (with a reference wavelength of 630 nm) to obtain a sample signal (i.e., OD570-OD630 for normalization data). The cytotoxicity of NAC was measured using a lactate dehydrogenase (LDH) cytotoxicity detection kit (TaKaRa Bio, Kyoto, Japan) according to the manufacturer’s protocol. Briefly, after NAC treatment, the supernatant was harvested and centrifuged at a maximum speed for 10 min. To determine the LDH activity in the supernatant, 100 μL supernatant was transferred into the wells of an optically clear 96-well plate, and reaction mixtures containing the catalyst and dye solution (Lod tetrazolium chloride and sodium lactate) were added into each well. The cells were then incubated for 30 min at 15–25 °C in the dark. Absorbance was measured at 490 nm, and the background control was measured at 600 nm using a microplate reader (BioTek, Winooski, VT, USA). The control group was defined as 100%.

### 2.6. Measurement of Reactive Oxygen Species (ROS) Level

To quantify intracellular ROS production, the cells were incubated for 30 min at 37 °C in serum-free RPMI 1640 medium containing 10 μmol/L 2′,7′-dichlorofluorescein diacetate (DCF-DA) (Thermo Fisher Scientific). The cells were washed with PBS, and the DCF-DA intensity in the cells was immediately measured at 495 nm (excitation)/529 nm (emission) using a microplate reader. ROS production in the cells was represented as a percentage of DCF-DA intensity relative to cell viability in each well, which was defined as 100%.

### 2.7. Intracellular Glutathione (GSH) and Nicotinamide Adenine Dinucleotide Phosphate (NADP^+^)/Reduced Form (NADPH) Levels

Total cellular GSH and oxidized GSH (glutathione disulfide, GSSG) were measured using a GSH fluorometric assay kit (BioVision; Milpitas, CA, USA). Cells were prepared via deproteinization; subsequent reagents were added, following the manufacturer’s instructions; and fluorescence was measured at excitation/emission = 340/420 nm. Fluorescence values from the standards and the samples were subtracted from the blank control values, and the GSH and GSSG contents were calculated via interpolation from the standard curve. Results were normalized to 10^6^ cells [35]. NADP^+^ and NADPH levels were analyzed using an NADP^+^/NADPH Quantification Colorimetric Kit (BioVision, Milpitas, CA, USA). Briefly, cells were washed with cold PBS and lysed with 800 μL NADP^+^/NADPH extraction buffer and were treated according to the manufacturer’s instructions. After centrifugation, the supernatant was collected for the measurement of NADP^+^/NADPH using a UV spectrophotometer at 450 nm.

### 2.8. Measurement of Cytokine Levels

At 12 h after APM and DEP stimulation, the levels of secreted IL-1β, IL-6, and TNF-α were measured in the cell culture supernatants. Quantification was performed using the respective ELISA kits (Invitrogen, Waltham, MA, USA), according to the manufacturer’s instructions.

### 2.9. RNA Extraction and Quantitative Real-Time Polymerase Chain Reaction (qRT-PCR)

Total RNA was extracted using the RNeasy Mini Kit (Qiagen, Venlo, Netherlands), and cDNA was synthesized from total RNA using the GoScript™ Reverse Transcription System (Promega, Madison, WI, USA). The cDNA was amplified with the QuantStudio5 real-time PCR system, using SYBR Green PCR Master Mix (Applied Biosystems, Foster City, CA, USA). The PCR primers used are listed in Table 1. The target gene chemokine (C-S-C motif) ligand (CXCL)-1, monocyte chemoattractant protein (MCP)-1, and NADPH dehydrogenase (quinone) (NQO)-1 expression levels were normalized to that of β-actin, calculated using the 2^−ΔΔCt^ method, and expressed as fold changes. The normalized value of the target gene expression level in the control group was set as 1.

### 2.10. Western Blotting Analyses

Cells were lysed in radioimmunoprecipitation assay buffer (Thermo Fisher Scientific) with a protease inhibitor cocktail. Protein concentrations were determined using Bradford reagent (Bio-Rad, Hercules, CA, USA). Proteins were separated using SDS-PAGE at 120 V for 90 min, then transferred to a polyvinylidene difluoride membrane (Bio-Rad) at 250 mA for 90 min via wet transfer. Nonspecific binding was blocked by incubating the membrane in 5% non-fat dry milk in Tris-buffered saline with Tween 20 (25 mmol/L Tris (pH 7.5), 150 mmol/L NaCl, and 0.1% Tween 20) for 1 h, followed by an overnight incubation at 4 °C with antibodies against anti-phospho-c-Jun N-terminal kinase (JNK), anti-JNK, anti-phospho-extracellular signal-regulated kinase (ERK), anti-ERK, anti-phospho-p38, anti-p38 (all from Abcam, Cam-bridge, MA), toll-like receptor (TLR) 4 (Invitrogen, Carlsbad, CA, USA), and β-actin (Santa Cruz Biotechnology, Santa Cruz, CA, USA). β-actin was used as the internal reference. Horseradish peroxidase-conjugated anti-rabbit or anti-mouse IgG (Cell Signaling Technology, Beverly, MA, USA) was used to detect antibody binding, which was visualized using an iBright CL1000 imaging system (Thermo Fisher Scientific) after treatment with an enhanced chemiluminescence reagent (Thermo Fisher Scientific). The densitometric analysis of each band was performed using Bright CL1000 image software (Thermo Fisher Scientific). For the quantification of specific bands, a square of the same size was drawn around each band for density measurement, and the value was adjusted to the background density near that band. The results were expressed as the relative ratio of the target protein to the reference protein, with the relative ratio of the target protein of the control group set to 1.

### 2.11. Statistical Analysis

Statistical analyses were performed using GraphPad Prism 8 software (GraphPad Software Inc., San Diego, CA, USA). Data are expressed as mean ± SD. The Shapiro–Wilk test was used to assess the normality of the data. A one-way analysis of variance was performed, followed by the Tukey or Dunnett test to perform statistical comparisons. The comparisons between the two groups were performed using the t-test for paired variables, or the Mann–Whitney U test for unpaired variables. The statistical significance was set at *p* < 0.05.

## 3. Results

### 3.1. Characterization of DEP and APM

Table 2 presents the characterization of APM and DEP. The DEP and APM particle sizes were less than 1 μm in size (mostly <600 nm) and about 30 nm, respectively [36]. DEP was also roughly spherical and was composed of small irregular agglomerates [36]. APM exhibited typical diesel soot particle morphologies, with nearly spherical and irregular carbonaceous particles [31]. In the Raman spectra of DEP, a D peak at 1340 cm^−1^ and a G peak at 1600 cm^−1^ is visualized [37]. Similar to DEP, in the Raman spectra of APM, two typical overlapping peaks are commonly shown [31,36]. Moreover, we measured the OC/EC ratio values of DEP and APM. OC and EC in PM are widely known for their toxicity [38], and they are linked to adverse effects on human health [39]. In this study, the OC/EC ratio value of APM was similar to the level of DEP. These physical characterizations and the OC/EC ratio of DEP and APM were commonly observed, and we used DEP and APM as PM.

### 3.2. PM Induces Oxidative Stress-Mediated Cell Damage in AMs

Exposure to PM induces cellular damage in the oxidative stress-mediated respiratory immune system [40,41,42,43]. First, we measured cell viability to determine the concentrations of PM, such as APM and DEP, at which cell damage occurred. Our results showed that PM, such as APM and DEP, significantly reduced cell viability from a minimum concentration of 100 μg/mL in AMs (Figure 1A); therefore, we selected 100 μg/mL APM and DEP. Morphological examinations conducted under an optical microscope indicated that several concentrations of APM/DEP induced AM death in 6 h (Figure 1B). Next, to determine whether PM-induced cell damage was mediated via oxidative stress, we evaluated cytotoxicity and ROS production via an LDH activity assay and DCF-DA fluorescence intensity measurement, respectively, after NAC treatment in PM-stimulated AMs. Compared with the untreated control, the LDH level gradually increased within AMs with increasing concentrations of NAC. We selected the non-toxic concentration of NAC at 10 mM and confirmed that PM induced cell damage by mediating ROS production in AMs, which was mitigated by NAC. Our results showed that PMs, such as APM and DEP, induce ROS-mediated cytotoxicity in AMs, and that NAC significantly inhibited the PM-induced increase of ROS production in AMs (Figure 2).

### 3.3. Effect of Coconut Oil on Cytotoxicity and Oxidative Stress in PM-Stimulated AMs

To investigate the effect of coconut oil on cytotoxicity, and its antioxidant activities in PM-stimulated AMs, we determined the concentrations of DW as a vehicle control, and of coconut oil without cytotoxicity. Our results showed that there was no change in cell viability up to 10% DW in complete medium (Figure 3A). Since there was no change in cell viability, up to 0.05% coconut oil in coconut oil-treated AMs (Figure 3B), and 0.0125% and 0.025% concentrations were chosen, for testing the efficacy of coconut oil in the *in vitro* study. Pretreatment with coconut oil reversed the PM-induced decrease in cell viability and the increase in oxidative stress in PM-stimulated AMs (Figure 3C,D). Furthermore, the mRNA level of NQO-1, an oxidative stress gene, was significantly increased within AMs by PM stimulation. Coconut oil pretreatment significantly decreased the PM-induced increase of NQO-1 mRNA levels (Figure 3E). Moreover, we confirmed that the PM-induced decrease of GSH levels was increased by coconut oil pretreatment in AMs (Figure 3F). The GSSG level, the ratio of GSSG-to-GSH, and the NADPH content, as indicators of cellular oxidative stress, were also reduced in PM-stimulated AMs pretreated with coconut oil (Figure 3G–I). These results indicate that PM, such as APM and DEP, induce oxidative stress-mediated cell damage in AMs, and that coconut oil can alleviate oxidative stress-mediated responses by upregulating GSH levels in PM-stimulated AMs.

### 3.4. Coconut Oil Suppresses Inflammatory Genes and Protein Expression in PM-Stimulated AMs

Coconut oil has anti-inflammatory activity against various diseases [25,26,27,28,29,30]. To examine whether coconut oil could regulate inflammatory responses in PM-stimulated AMs, we evaluated the levels of inflammatory genes, such as MCP-1 and CXCL-1, and inflammatory proteins, such as TNF-α, IL-1β, and IL-6, using qRT-PCR and ELISA assays, respectively. Our results showed that APM and DEP stimulation significantly increased the levels of MCP-1 and CXCL-1 genes, as well as those of the TNF-α, IL-1β, and IL-6 proteins in AMs. These levels were significantly reduced when treatment with coconut oil was used (Figure 4). These findings indicate that coconut oil potently inhibits inflammatory responses via the regulation of PM-induced inflammatory mediators in AMs.

### 3.5. Coconut Oil Attenuates PM-Induced MAPK Signaling Activation in AMs

PM has been reported to activate intracellular MAPK signaling pathways and induce metabolic changes in human bronchial epithelial cells and mouse macrophages [17,44,45]. MAPK is an essential regulator of inflammatory mediators, including proinflammatory cytokines [45,46]. Hence, to evaluate the effect of coconut oil on MAPK signaling, we investigated the phosphorylation levels of JNK, p38, and ERK in APM– or DEP–exposed AMs and in APM– or DEP–induced AMs pretreated with coconut oil. Our results showed that phospho-JNK, total JNK, phospho-p38, and phospho-ERK expression were significantly increased within APM– and DEP–induced AMs compared with vehicle groups (Figure 5). Pretreatment with coconut oil significantly decreased APM– or DEP–induced MAPK signaling. These results indicate that coconut oil may alleviate the inflammatory response via the regulation of MAPK signaling in PM-induced AMs.

### 3.6. Coconut Oil Inhibits PM-Induced TLR4 Activation in AMs

TLR4 signaling plays an important role in the regulation of PM-induced inflammatory responses [47]. We investigated whether coconut oil regulated TLR4 signaling in PM-stimulated AMs, using Western blotting analysis. The results showed that PM stimulation significantly upregulated TLR4 expression in AMs, and this effect was significantly abrogated by coconut oil pretreatment (Figure 6).

## 4. Discussion

Our results, for the first time, demonstrate that coconut oil, a natural extract and functional food, attenuates oxidative stress-associated inflammatory responses via the modulation of TLR4 and MAPK activation in PM-stimulated AMs.

PM leads to health risks posed by pollutants from the ambient air or the workplace [41,48]. DEP is a mixture of various chemical components, including EC and OC, which constitute the carbon core, metals, and trace elements; organic compounds such as alkanes, branched alkanes, alkyl-cycloalkanes, alkyl-benzenes, and polycyclic aromatic hydrocarbons (PAHs); and various cyclic aromatics [49]. PAHs and related compounds, such as quinones, are metabolized to even more toxic metabolites upon ingestion [50], primarily because of their redox potential and their ability to induce oxidative stress [51]. To date, many researchers have studied in humans the toxicity of DEP as one of the natural PM components, and have reported that DEP is associated with acute irritation and the induction of respiratory symptoms. Consequently, it could exacerbate pulmonary diseases in response to known allergens and pathogens [52,53]. However, it is difficult to determine the key soot factors within a mixture of inorganic compounds, trace metals, and PAHs that play a major role in inducing toxicity [54]. Hence, we generated an APM from a single origin source as ultrafine particles. In a previous study, the APM generated in our lab maintained its original physical properties and possessed a chemically controllable surface [31]. ROS-mediated cytotoxicity is significantly increased via an increase of oxygenated functional groups on the surface of APM in epithelial cells [31]. Thus, we selected DEP and APM as PM. 

AMs are responsible for the first line of defense against pathogens and pollutants, which can activate the innate immune response in the lungs. AMs participate in host defense by engulfing and digesting pathogens, and they maintain clear air spaces by engulfing foreign materials and minimizing the exposure of other airway cells to these substances [11,12]. This immune cell plays an essential role in maintaining a balance between the defense against pathogens and the tolerance towards innocuous stimuli [55]. AMs are the predominant cells in the lungs that are responsible for the processing and removal of inhaled particulate matter. When exposed to atmospheric particles, activated AM cells may release excessive amounts of mediators, such as oxygen radicals, proteases, and growth-regulating proteins, which could then be involved in the pathogenesis of acute and chronic lung diseases [56]. 

Coconut oil is used as a health food and has nutritional effects. *In vitro* and *in vivo* studies have reported that coconut oil exhibits various biological activities, including antioxidant, anti-inflammatory, anticancer, analgesic, antimicrobial, and antipyretic properties [23,25,26]. In fact, despite many studies, little has been reported about the effects of coconut oil on lung diseases. In an *in vivo* study, Luiz Henrique C reported that coconut oil prevented airway tissue from exhibiting peribronchial inflammatory infiltrate, epithelial hyperplasia, and smooth muscle thickening in guinea pigs that were submitted to ovalbumin sensitization [27]. However, there have been no reports on the biological effects of coconut oil from exposure to PM. Hence, we aimed to assess the protective effects of coconut oil in PM-stimulated AMs, focusing on its antioxidant and anti-inflammatory activities, and investigate its mechanism of action.

In this study, to investigate the protective effect of coconut oil in PM-stimulated AMs, we first optimized the concentration of APM and DEP by treating AMs with different concentrations, and examined the cell viability of PM-stimulated AMs. Researchers have shown that DEP induces the cytotoxicity of cell death over large ranges (50–3000 μg/mL) [57,58]. Therefore, we confirmed cytotoxicity by stimulating DEP or APM within those ranges in AMs, to find the optimal concentration. We found that APM and DEP induced early cytotoxicity at 100 μg/mL in AMs. We also confirmed that PM-induced ROS production and cytotoxicity in AMs are regulated by NAC treatment. Therefore, we examined the protective effect of coconut oil on oxidative activity by measuring cytotoxicity, ROS production, and GSH activity in PM-stimulated AMs. GSH plays a role in preventing damage to important cellular components caused by reactive oxygen species, such as free radicals, peroxides, and lipid peroxides, and it is associated with hypoxia, lung diseases, and aging [40]. In cells, GSH is oxidized during peroxide disposal by GSH peroxidase, and GSSG is reduced to two molecules of GSH, with reducing equivalents from the coenzyme NADPH [59]. The relative amounts of reduced (GSH) and oxidized GSH (GSSG) serve as markers of oxidative stress; thus, the GSSG/GSH ratio increases under oxidative stress [60]. In this study, our results showed that 100 μg/mL PM decreased cell viability and GSH levels, increased GSSG content, increased the GSSH/GSH ratio, increased NADPH level, and increased ROS production in AMs. The PM-induced changes were recovered by coconut oil in the AMs. Moreover, we measured via qRT-PCR the levels of NQO-1, an oxidative stress gene, in PM-stimulated AMs. PM-induced NQO-1 mRNA levels were significantly decreased by coconut oil in AMs. These results indicate that oxidative stress is a key modulator and initiator of PM-induced cell damage, and that coconut oil has potent antioxidant properties [22,24,26] in PM-stimulated AMs. Therefore, we suggest that coconut oil prevents cell damage by regulating GSH activity during PM-induced oxidative stress in AMs. 

Another important function of coconut oil is its anti-inflammatory activity. We tested the levels of various inflammatory mediators in PM-stimulated AMs by qRT-PCR and Western blotting. Our results showed that PM upregulated the mRNA levels of MCP-1 and CXCL-1, as well as the protein levels of TNF-α, IL-6, and IL-1β in AMs. Coconut oil downregulates various PM-induced inflammatory mediators in AMs. In the respiratory system, AMs clearly maintain the air space between cells in the respiratory system by the phagocytosis of pathogens and debris, and foreign material-phagocytosed AMs release the prime immune responses of mediators such as TNF-α, IL-1β, IL-6, and neutrophil chemoattractants [60]. TNF-α, IL-1β, and IL-6 are widely studied pro-inflammatory cytokines that play a key role in the pathogenesis of numerous inflammatory diseases, and in the host defense mechanism [61]. In addition, the levels of these inflammatory cytokines are more sensitive to PM in COPD patients [62]. Several studies have demonstrated that tissue-resident macrophages produce chemoattractants such as CXCL-1 and MCP-1 [63,64] after engulfing particles. The chemokine CXCL-1 plays an important role in inflammation, as it is a major chemoattractant that is responsible for recruiting neutrophils. MCP-1 has been reported to promote macrophage and T-lymphocyte infiltration after tissue injury [65]. These results indicated that coconut oil can prevent PM-induced early inflammatory responses in AMs. 

To elucidate the mechanism underlying the biological activities of coconut oil in APM– and DEP–stimulated AMs, we investigated the MAPK pathways. The MAPK signaling pathway is one of the most critical intracellular signal transduction systems, and it is generally activated by a broad array of extracellular or intracellular stimuli [20]. In this study, we found that the phosphorylation levels of p38, JNK, and ERK in AMs were higher than those in the control groups, using Western blotting analysis. Nevertheless, the PM-upregulated MAPK pathways were downregulated after pretreatment with coconut oil. TLR4 is an important upstream factor in the MAPK pathway, and it contributes to immune responses by regulating the levels of TNF-α, IL-1β, IL-6, inducible nitric oxide synthase, and nitric oxide via the MAPK pathway [66,67]. In this study, we showed that PM-induced TLR4 expression was significantly decreased by coconut oil in AMs. Moreover, we examined Akt signaling (Appendix A), an upstream factor of the nuclear factor kappa B (NF-κB) pathway in PM-stimulated AMs. In our previous study, we showed that NF-κB levels in nuclear protein extracts from AMs were substantially increased within DEP–stimulated AMs [9]. Consistent with these observations, we showed that PM exposure induces Akt phosphorylation in APM– and DEP–stimulated AMs. In conclusion, this study demonstrated that coconut oil can be potentially developed as an adjuvant therapy for PM-induced oxidative stress-mediated lung inflammation. This research is limited by the differences between APM, DEP, and natural atmospheric PM. We can synthesize APM with varying physical and chemical properties through temperature control. Therefore, it is necessary to evaluate the efficacy of coconut oil under various conditions. Future studies aim to investigate the potential efficacy of coconut oil in an *in vivo* study.

## 5. Conclusions

As a result of our *in vitro* study, we found that coconut oil had protective effects on oxidative stress and inflammatory responses that are induced by PM in AMs. Coconut oil has been shown to reduce APM/DEP–induced inflammation by modulating the TLR4/MAPK pathway in AMs, as well as by conferring protection against adverse respiratory effects caused by PM exposure. It has been demonstrated in this study that coconut oil can potentially be used as an adjuvant therapy against PM-induced oxidative stress-mediated lung inflammation.

## Figures and Tables

**Figure 1 molecules-27-02898-f001:**
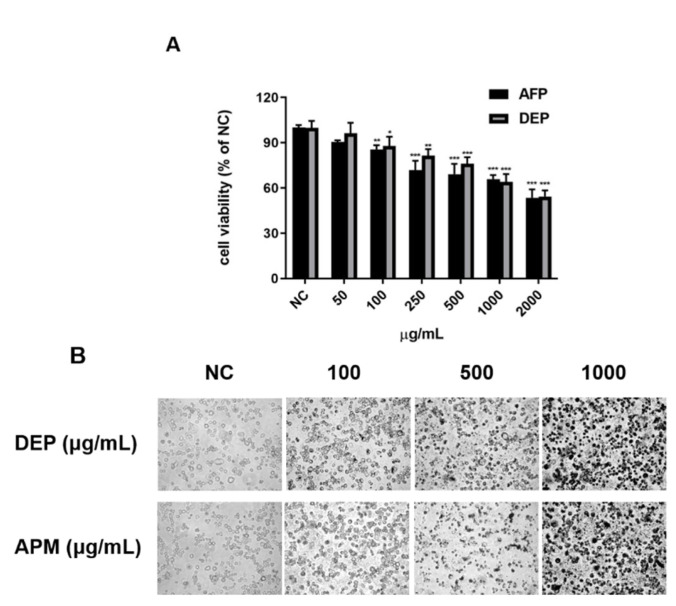
Cytotoxicity of artificial particulate matter (APM) and diesel exhaust particles (DEP) in alveolar macrophages (MH-S). Negative controls (NC) are untreated samples. MH-S was treated with APM and DEP of different concentrations for 6 h. (**A**) Cell viability was measured using the 3–(4,5–dimethylthiazol–2–yl)–2,5–diphenyltetrazolium bromide (MTT) assay in the APM– and DEP–stimulated MH-S cells. (**B**) Morphological changes in MH-S by exposure to APM/DEP. Data are presented as the mean ± SD (n = 6 per group). * *p* < 0.05, ** *p* < 0.01, or *** *p* < 0.001 vs. NC.

**Figure 2 molecules-27-02898-f002:**
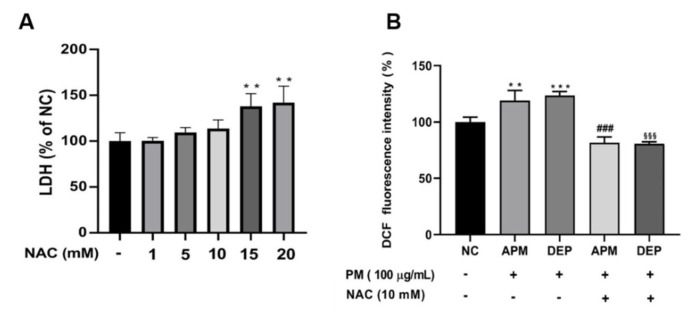
Artificial particulate matter (APM) and diesel exhaust particles (DEP) induce oxidative stress in alveolar macrophages (MH−S). (**A**) Cytotoxicity of N−Acetyl Cysteine (NAC). (**B**) Reactive oxygen species level was measured using 2′,7′−dichlorofluorescein diacetate (DCF−DA) staining. Negative controls (NC) are untreated samples. MH−S was pretreated with NAC for 3 h and was stimulated with 100 μg/mL APM or DEP for 3 h. Data are presented as the means ± SD (n = 4 per group). ** *p* < 0.01, or *** *p* < 0.001 vs. NC; ^###^ *p* < 0.001 vs. APM; ^§§§^ *p* < 0.001 vs. DEP group.

**Figure 3 molecules-27-02898-f003:**
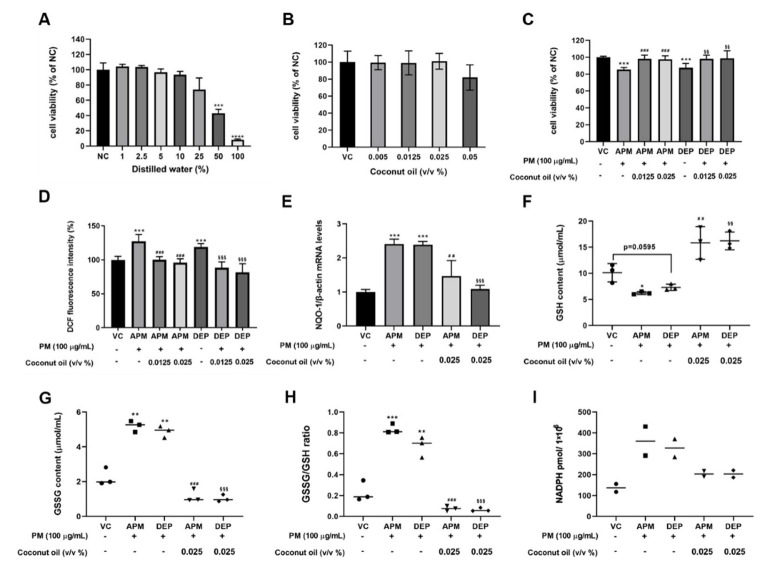
Effects of coconut oil on oxidative stress in artificial particulate matter (APM) and diesel exhaust particles (DEP)−stimulated alveolar macrophages (MH−S). (**A**,**B**) Cytotoxicity assessment of distilled water (DW) and coconut oil the 3–(4,5–dimethylthiazol–2–yl)–2,5–diphenyltetrazolium bromide (MTT) assay. (**C**) Cytotoxicity assessment of coconut oil in APM and DEP−stimulated MH−S. (**D**) Reactive oxygen species (ROS) production was measured using 2′,7′−dichlorofluorescein diacetate (DCF−DA) staining. Data represent the mean ± SD (n = 8 per group). (**E**) NADPH dehydrogenase (quinone) (*NQO1*) gene level was measured using real−time PCR. (**F**) Glutathione (GSH), (**G**) glutathione disulfide (GSSG), (**H**) GSSG/GSH ratio, and (**I**) Nicotinamide adenine dinucleotide phosphate (NADPH) levels in MH−S. Negative controls (NC) are untreated samples. Vehicle control (VC) was treated with cell medium including 10% DW. Data are presented as the means ± SD of three independent experiments. * *p* < 0.05, ** *p* < 0.01 or *** *p* < 0.001, **** *p* < 0.0001 vs. NC or VC; ^##^ *p* < 0.01 or ^###^ *p* < 0.001 vs. APM group; ^§§^ *p* < 0.01 or ^§§§^ *p* < 0.001 vs. DEP group.

**Figure 4 molecules-27-02898-f004:**
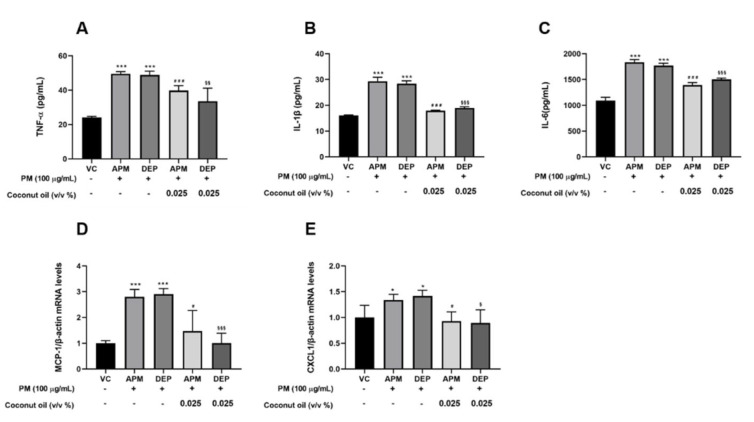
Effects of coconut oil on artificial particulate matter (APM) and diesel exhaust particles (DEP)−induced inflammation in alveolar macrophages (MH−S). Protein levels of (**A**) tumor necrosis factor−α (TNF−α), (**B**) interleukin (IL)−1β, and (**C**) IL−6 in MH−S were determined using enzyme−linked immunosorbent assay (ELISA). (**D**,**E**) Relative gene expression of inflammatory factors in APM/DEP–stimulated MH−S cells. Vehicle control (VC) was treated with cell medium including 10% distilled water. All experiments were independently performed in triplicate. * *p* < 0.5 or *** *p* < 0.001 vs. VC; ^#^ *p* < 0.05 or ^###^ *p* < 0.001 vs. APM; ^§^ *p* < 0.05 or ^§§^ *p* < 0.01, ^§§§^ *p* < 0.001 vs. DEP.

**Figure 5 molecules-27-02898-f005:**
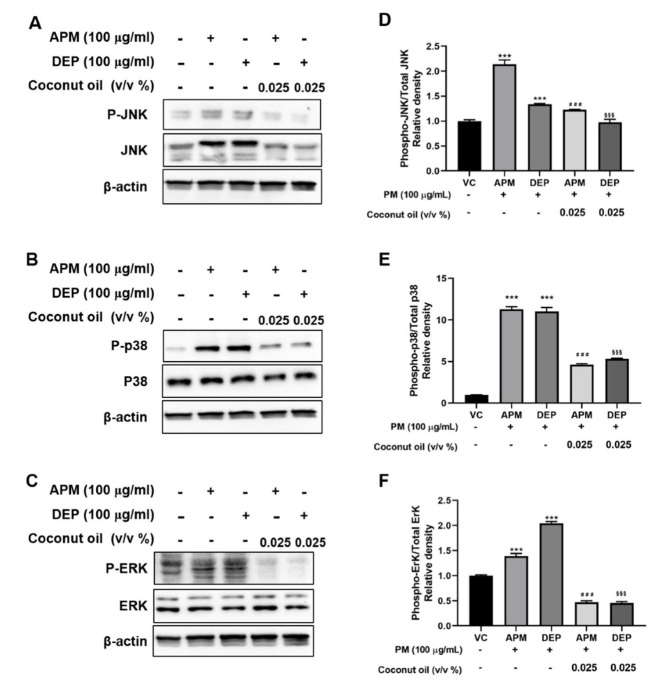
Effect of coconut oil on mitogen−activated protein kinase (MAPK) pathway in artificial particulate matter (APM) and diesel exhaust particles (DEP)−stimulated alveolar macrophages (MH−S). (**A**–**C**) Phosphorylation of extracellular receptor-activated kinase (ERK), C−Jun N−terminal kinase (JNK), and p38 MAPK was detected via Western blotting. (**D**–**F**) Relative densities of p−JNK, p−p38 MAPK, and p−ERK proteins. Vehicle control (VC) was treated with cell medium including 10% distilled water. Data are presented as the means ± SD (n = 3 per group). The normalized value of target protein expression level in the control group was set to 1. *** *p* < 0.001 vs. VC; ^###^ *p* < 0.001 vs. APM; ^§§§^ *p* < 0.001 vs. DEP groups.

**Figure 6 molecules-27-02898-f006:**
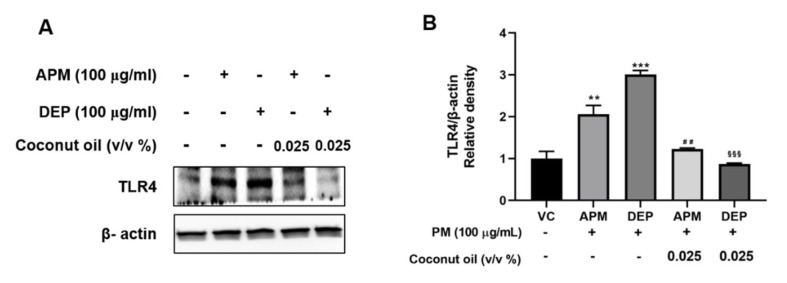
(**A**) Representative Western blots and (**B**) relative density of toll−like receptor (TLR) 4 expression in artificial particulate matter (APM) and diesel exhaust particles (DEP)−stimulated alveolar macrophages (MH−S). Vehicle control (VC) was treated with cell medium including 10% distilled water. Data are presented as the means ± SD (n = 3 per group). ** *p* < 0.01 or *** *p* < 0.001 vs. VC; ^##^ *p* < 0.01 vs. APM group; ^§§§^ *p* < 0.001 vs. DEP group.

**Table 1 molecules-27-02898-t001:** The forward and reverse primer.

Gene Name	Forward Primer	Reverse Primer
* **CXCL–1** *	5′-AACCGAAGTCATAGCCACAC-3′	5′-GTTGGATTTGTCACTGTTCAGC-3′
* **MCP–1** *	5′-TGATCCCAATGAGTAGGCTGGAG-3′	5′-ATGTCTGGACCCATTCCTTCTTG-3′
* **NQO–1** *	5′-GGTGATATTTCAGTTCCCATTG-3′	5′-ACTCTCTCAAACCAGCCTTTC-3′
* **β–Actin** *	5′-GGCACCACACCT TCTACAATG-3′	5′-GGGGTGTTGAAGGTCTCAAAC-3′

**Table 2 molecules-27-02898-t002:** Characterization of DEP and APM.

	DEP ^(a)^	APM
**Particle size (nm)**	mostly <600 nm [36]	approximately 30 nm [31]
**Pore diameter (nm)**	4−35 nm ^(b)^	approximately 30 nm [31]
**Morphology**	small irregular agglomerates and roughly spherical [36]	spherical and irregular carbonaceous particles [31]
**Raman spectrum**	D peak: 1340, G peak: 1600 [37]	D peak: 1340, G peak: 1600 [31]
**OC/EC ratio values ^(c)^**	1.9 ± 0.001	2.2 ± 0.001

^(a)^ Diesel exhaust particles, DEP (SRM2975); ^(b)^ Certified pore diameter in SRM2975; ^(c)^ Ratios of organic carbon (OC)/elemental carbon (EC).

## Data Availability

Not applicable.

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
