# Peer review of "Coconut Oil Alleviates the Oxidative Stress-Mediated Inflammatory Response via Regulating the MAPK Pathway in Particulate Matter-Stimulated Alveolar Macrophages"

_molecules, 2022, doi:10.3390/molecules27092898_

Round 1
Reviewer 1 Report
Thank you for granting me the opportunity to review this manuscript. In this piece of work, Chen et al. investigated the toxicity effect of artificial particulate matter and diesel particulate matter on alveolar macrophages (in vitro). The authors additionally assessed the protective role of coconut oil on particulate matter-induced oxidative stress and inflammatory responses in alveolar macrophages and investigated its potential mechanism of action.
Kindly find below my comments for your response.
Abstract
Line 15: revise to “……..demonstrates antioxidant and anti-inflammatory properties…”
Line 16: add “on” before “the”
Line 17: correct “especial-ly” with “especially”
Line 22 and 23: Kindly, indicate the p values
Line 27: correct “re-versed” to “reversed”
Introduction
Line 46-48: Kindly provide references for that statement
Line 62: Introduce “demonstrating” before “several”
Materials and Methods
This section should come after the “Introduction”.
Line 323: correct “manufactur-er’s” with “manufacturer’s”
Line 333: correct “me-dium” with “medium”
Line 336: correct “meas-ured” to “measured”
Line 343: correct “manufac-turer’s” to “manufacturer’s”
Line 349: Expand the abbreviation “PBS”
Line 355: correct “accord-ing” to “according”
Line 373: correct “inhibi-tor” to “inhibitor”
Line 378: correct “fol-lowed” to “followed”
Statistical analysis
The authors should indicate if data normality was checked for the data prior to analysis.
Results
Line 152: correct “media-tors” to “mediators”
All the Figures must always stand alone. The authors must expand all “abbreviations” used in the Figures. For example, in Figure 1, the authors must expand the abbreviation “NC”. The VC in Figure 1 must also be expanded.
All the abbreviations in Figure 2 must also be expanded.
In Figure 3, abbreviations including “NC” and “VC” must all be expanded.
Discussion
Line 184: correct “met-als” to “metals”
Line 208: The first part deals with a description of what was done. The authors should rather discuss the results.
Line 249: The authors consistently state what they did. That has already been presented in the “Materials and Method”. They should rather “discuss” the results.
General comments
- In the Methods employed by the authors for some of assessment of the parameters, consistently I see that the authors make no reference to any references in literature. Does that mean the authors are the first to use the procedures employed? Some other researchers have undertaken some of the analysis undertaken and thus reference to their procedures may be helpful.
- In the Discussion section, I see that, rather than the authors zooming straight to discuss the results, they go about talking about what they did before. That looks like a repetition of the “Methods”. The authors should rather discuss their findings in the context of what they or other researchers have done in the past.
- Concerning the statistical analysis, there were places like in Figures 3 B for example where the authors have prepared varying concentrations (more than 2) of coconut oil emulsion formulations and investigated their efficacy. In that instance the use of ANOVA will be appropriate rather than using the t test which is ideal for checking statistically significant difference in means of two samples. Also, at Line 304 to 306, the authors stated this “To optimize the treatment concentration, cells were treated with 304 varying concentrations of APM and DEP (50, 100, 250, 500, 1000, and 2000 μg/mL), coconut oil (0.005, 0.0125, 0.025, and 0.05%), and NAC (1, 5, 10, 15, and 20 mM)”. The statistical analyses that can be done will require ANOVA and not a t-test. This must be made clear in the “Statistical Analysis section”.
- The authors must expand all the abbreviations used in all the Figures presented under the result section.
- The authors must re-arrange the sequence of the presentation of the work. It must be in this order “Introduction, Materials and Methods, Statistical Analysis, Results, Discussion and Conclusion”.
Reviewer 2 Report
This paper discusses the PM-induced inflammatory response and the amelioration of this inflammatory response by coconut oil. This is a very interesting and meaningful experiment. However, there are still many problems in this paper that need to be revised. I think this paper should be subjected to a major revision before being published. Details are listed as follows:
- There are many abbreviations in the manuscript, it is recommended to add an abbreviation description list.
- In this study, the particle parameters of APM and DEP are an important premise and foundation of this study. However, there is no relevant characterization of these particles. Theses data should be supplemented.
- What is the selection basis for the concentrations of APM and DEP particles?
- Are cellular models suitable as models for PM-induced inflammatory responses? Is there a relevant basis for the selection of this model to reflect PM-induced diseases? I think this should be introduced in the introduciton section.
Round 2
Reviewer 1 Report
Figures must stand alone. Consequently, abbreviations including APM, DEP, APM and DEP and all other unexpanded abbreviations must be expanded.
Reviewer 2 Report
Accept.
Author Response
Thank you for your good response.